# Sex-dependent and -independent transcriptional changes during haploid phase gametogenesis in the sugar kelp *Saccharina latissima*

**Gareth A. Pearson**[1,☉]*, **Neusa Martins**[1,☉], **Pedro Madeira**[1], **Ester A. Serrão**[1], **Inka Bartsch**[2]

**1** Centre for Marine Sciences (CCMAR)-CIMAR, University of Algarve, Portugal, **2** Alfred-Wegener-Institute, Helmholtz Center for Polar and Marine Research, Am Handelshafen, Germany

☉ These authors contributed equally to this work.
* gpearson@ualg.pt

**Data Availability Statement:** The clean read data for female and male gametophyte samples analysed in this paper are available in the Sequence Read Archive (SRA; accessions SRX6058950-

## Abstract

In haplodiplontic lineages, sexual reproduction occurs in haploid parents without meiosis. Although widespread in multicellular lineages such as brown algae (Phaeophyceae), haplodiplontic gametogenesis has been little studied at the molecular level. We addressed this by generating an annotated reference transcriptome for the gametophytic phase of the sugar kelp, *Saccharina latissima*. Transcriptional profiles of microscopic male and female gametophytes were analysed at four time points during the transition from vegetative growth to gametogenesis. Gametogenic signals resulting from a switch in culture irradiance from red to white light activated a core set of genes in a sex-independent manner, involving rapid activation of ribosome biogenesis, transcription and translation related pathways, with several acting at the post-transcriptional or post-translational level. Additional genes regulating nutrient acquisition and key carbohydrate-energy pathways were also identified. Candidate sex-biased genes under gametogenic conditions had potentially key roles in controlling female- and male-specific gametogenesis. Among these were several sex-biased or -specific E3 ubiquitin-protein ligases that may have important regulatory roles. Females specifically expressed several genes that coordinate gene expression and/or protein degradation, and the synthesis of inositol-containing compounds. Other female-biased genes supported parallels with oogenesis in divergent multicellular lineages, in particular reactive oxygen signalling via an NADPH-oxidase. Males specifically expressed the hypothesised brown algal sex-determining factor. Male-biased expression mainly involved upregulation of genes that control mitotic cell proliferation and spermatogenesis in other systems, as well as multiple flagella-related genes. Our data and results enhance genome-level understanding of gametogenesis in this ecologically and economically important multicellular lineage.

SRX6058965) under BioProject accession PRJNA547989. The Transcriptome Shotgun Assembly project has been deposited at DDBJ/EMBL/GenBank under the accession GHNM00000000. The version described in this paper is the first version, GHNM00000000.1 (GHNM01000001:GHNM01034002).

**Funding:** This work was supported by the Portuguese Science Foundation (FCT; https://www.fct.pt/index.phtml.pt) programs UID/Multi/04326/2019, GENEKELP-PTDC/MAR-EST/6053/2014 and MARFOR-Biodiversa/0004/2015 (https://www.biodiversa.org/1019) to ES and GAP. The BiodivERsA project MARFOR includes the funding agencies Formas, Naturvardsverket, ANR, FCT, MEC, FRCT, DFG. Author NM is supported by a grant from FCT (SFRH/BPD/122567/2016) and ES by a Pew Marine Fellowship (https://www.pewtrusts.org/en/projects/marine-fellows). This work was additionally supported by a STSM Grant from the COST Action "Phycomorph" FA1406 (https://www.phycomorph.org) to NM. The funders had no role in study design, data collection and analysis, decision to publish, or preparation of the manuscript.

**Competing interests:** The authors have declared that no competing interests exist.

## Introduction

Gametogenesis is a fundamental process for sexual reproduction. In diploid organisms, the sequential activation of two major developmental programs are coordinated; expression changes causing a switch from mitotic cell division (i.e., growth) to reductive meiotic division producing haploid cells, and subsequent activation of the morphogenetic program to produce differentiated haploid (male and female) gametes [1]. However, haplodiplontic lineages deviate from this pattern, because sex is expressed in the haploid phase of the life cycle, thereby uncoupling gametogenesis from meiosis (see [2,3]). Haploid phase gametogenesis can thereby be viewed as a relatively simplified process, limited to the integration of gametogenic signals to fix cell-fate decisions and produce gametes.

Brown algae in the family Laminariales (kelps, *sensu stricto*) maintain an extremely heteromorphic haplodiplontic life cycle alternating between haploid microscopic gametophytes and diploid macroscopic sporophytes. Kelp sporophytes are the most developmentally complex and largest members of the Phaeophyceae, an independently evolved eukaryotic multicellular lineage [4]. In contrast, gametophytes are cryptic, either free-living or endophytic [5] with a highly simplified filamentous morphology specialised for gamete production. In addition to their unique evolutionary position on the tree of multicellular life, kelp gametophytes are an attractive model system for the study of gametogenesis. Gametophytes can be isolated and maintained in long-term vegetative culture (i.e., for decades; see [6,7]). Gametogenesis can be induced in the laboratory by blue light [8] or Fe addition [9], and development to fertilization is rapid under optimal conditions (ca. 10 days). Sexual development proceeds independently in male and female gametophytes (dioicy), with external fertilization by biflagellate sperm that are discharged and attracted in response to the release by eggs of the pheromone lamoxirene [10].

Although evidence for genetic sex determination has long existed [11], the UV sex determination system in *Ectocarpus* [12] and related brown algae including Laminariales [13] was confirmed only recently, as genomic resources for brown algae have become available [4, 14]. Thus, haploid sex chromosome evolution across a broad evolutionary range of brown algae has recently advanced considerably [13], reviewed by [15], and some studies have looked at gene expression underlying sexual dimorphism [16, 17]. However, information about gene expression changes during gametogenesis in haplodiplontic and oogamous kelps remains limited (but see [18]).

There are both ecological and economic rationales for studying the genetic and genomic underpinnings of kelp sexual development. The fundamental role of kelps in structuring marine forest ecosystems across cold to temperate regions of both hemispheres is increasingly threatened by climate change-induced range shifts, over-exploitation, and habitat destruction [19–23]. The sugar kelp *Saccharina latissima* is a highly successful amphi-boreal species [24], in which a diverse habitat range has promoted the development of distinct morpho- and ecotypes [24, 25] that may indicate a large reservoir of functional genetic variation with potential use in commercial breeding programs [26]. It also suggests that as range shifts gather pace, elements of this diversity could be lost.

To address this knowledge gap, we have undertaken the first transcriptional study of male and female gametogenesis in *S. latissima*. Our results provide evidence both for common elements expressed during the commitment to gametogenesis from the vegetative phase, as well as sex-specific developmental programs in males and females.

## Materials and methods

### Experimental culture conditions and sampling

Unialgal female and male gametophyte cultures of *Saccharina latissima*, each derived from the meiospores of a single sporophyte, were isolated from Oslo, Norway (AWI seaweed culture

collection: ♀ 3301, ♂ 3300) and Spitsbergen, Svalbard (AWI seaweed culture collection: ♀ 3124, ♂ 3123). They were maintained in a vegetative stage at 15°C in 3 μmol photons $m^{-2}$ $s^{-1}$ of red light (RL; LED Mitras daylight 150 controlled by ProfiLux 3, GHL Advanced Technology, Kaiserslautern, Germany) under a 16:8 h light:dark cycle in sterile full strength Provasoli enriched seawater (PES; [27]). Gametophyte vegetative growth was enhanced to have sufficient material for the gametogenesis induction experiment and RNA extraction; female and male gametophyte material from both strains was transferred to seawater enriched with $8 \times 10^{-4}$ M N and $1 \times 10^{-4}$ M P, at 17°C, under continuous irradiance of 6 μmol photons $m^{-2}$ $s^{-1}$ of RL for 6 weeks. The seawater medium was renewed weekly. The irradiance conditions chosen for vegetative gametophyte growth were optimal based on initial experiments indicating improved culture health (mortality, qualitative assessment of pigmentation) in RL compared to WL (pers. obs.). Nutrient conditions were adapted from [28].

### Induction of gametophyte reproduction

To induce gametogenesis, female and male gametophytes from both strains (biological replicates) were gently separated into small fragments (~1 mm in length) and cultured separately into Petri dishes (9.5 cm diameter, height 5.5 cm) with ½ strength PES at 10 °C. The irradiance was 15 μmol photons $m^{-2}$ $s^{-1}$ of white light (WL; LED MITRAS lightbars 150 Daylight, GHL Advanced Technology, Kaiserslautern, Germany) in a 16:8 h light:dark cycle. Male and female gametophytes were sampled for RNA (flash-frozen in liquid nitrogen and stored at -80 °C) after 1, 6 and 8 days in WL to follow the developmental stages of gametogenesis. Control gametophytes were sampled under RL growth conditions described above (0 days vegetative control). All samples were taken before gametophytes released eggs and sperm, based on observations of the cultures (Fig 1).

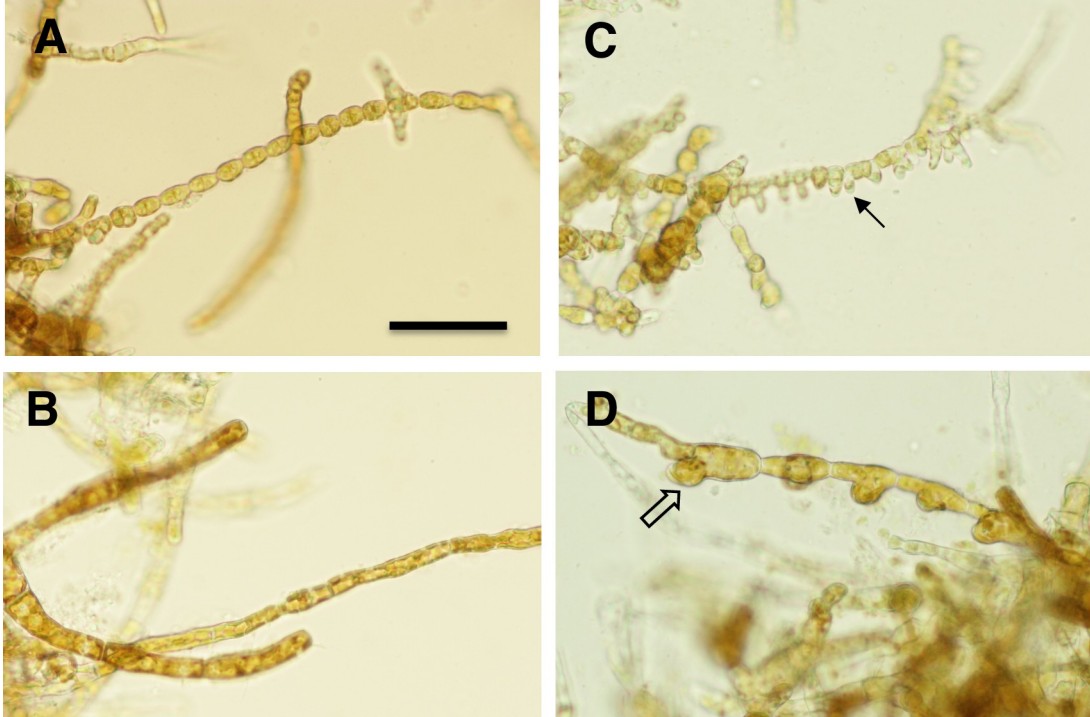

**Fig 1. Cultured *Saccharina latissima* gametophytes used in this study.** Vegetative filaments of A) males and B) females under RL vegetative growth conditions. The same male C) and female D) cultures after 8 d in WL gametogenic conditions. Solid and open arrows indicate sites of antheridial and oogonial development, respectively. Scale bar = 50 μm.

## RNAseq analysis and de novo reference transcriptome assembly

Total RNA was extracted from lyophilized tissue equivalent to between 100–200 mg FW male and female gametophyte culture per sample following established protocols [29] and sequenced by a service provider (100 bp paired-end Illumina HiSeq 4000; BGI, China).

The raw sequence data were evaluated with standard quality control tools (FastQC v0.11.7; https://www.bioinformatics.babraham.ac.uk/projects/fastqc/). Prior to *de novo* transcriptome assembly, quality-filtered male and female reads were digitally normalized to reduce redundancy, equalize *k*-mer coverage and remove rare *k*-mers potentially arising from sequencing errors [30]. Assemblies (Velvet-Oases; [31]) were built over the *k*-mer range 21–61 (step size of 10), and these 5 assemblies were subsequently merged with transfuse (https://github.com/cboursnell/transfuse). The merged transcriptome was queried by Diamond [32] in BLASTX mode against Stramenopile proteins (subset from NCBI nr); contigs with top blast hits against Phaeophyceae were retained, and the remaining contigs containing potential contaminants were removed from the analysis.

A proportion of contigs from brown algae *de novo* transcriptomes may be "polycistronic", likely resulting from brown algal genome structure, in which closely adjacent genes are transcribed from alternate strands [4]. Therefore, to produce the final reference transcriptome, putative open reading frames (ORFs) were identified with FragGeneScan [33], and clustered at 97% nucleotide identity with VSEARCH [34]. Sequences with length < 200 nt were discarded.

## Transcriptome completeness

We assessed the completeness of the final merged, Phaeophyceae-screened reference transcriptome with BUSCO v2.0 [35], by querying predicted proteins against the eukaryotic reference database, using online resources provided by CyVerse https://www.cyverse.org/.

## Differential expression analysis

High quality reads were mapped onto the reference transcriptome using the RSEM (v1.2.31) wrapper script and Bowtie2 [36, 37]. Expected count data were analysed in Bioconductor 3.8 using edgeR and limma [38, 39]. A total of 11,916 transcripts were retained for analysis after filtering for transcripts with > 4 counts per million (CPM, approximately 20 reads/transcript) in at least 6 samples. Samples from the two available strains of *S. latissima* (SLO—Oslofjord and SLS–Spitzbergen) were used as biological replicates to investigate transcriptome expression profiles in response to the factors "sex" (two levels; male and female [M and F]) and "time" (four levels; vegetative growth in RL [= day 0], and 1, 6 and 8 days following a transfer to WL to initiate gametogenesis). Differential expression (DE) was used to identify up- and down-regulated transcripts between groups defined by combinations of sex and time (false discovery rate [FDR] $\leq$ 0.05). The analysis was repeated on KEGG orthology (KO) terms, after summing transcript reads corresponding to unique KO entries.

Differentially expressed genes reported in this study were subject to additional confirmatory phylogenetic checks, since known pathogens and parasites of brown algae are themselves stramenopiles (oomycetes, labyrinthulids), while other phototrophic stramenopile lineages such as diatoms may be present in small numbers in culture. To do this, amino acid alignments were made (Muscle; [40]) from all blastx hits (Expect-value cutoff $\leq$ 1e$^{-10}$), alignments were trimmed with GBlocks [41] before phylogenetic tree construction (PhyML, LG model and aLRT branch support; [42]). Some suspect contigs sister to labyrinthulids or in very few cases diatoms were screened and removed from further analysis.

Heatmaps (ComplexHeatmap R package; [43]) were prepared from a matrix of all DE transcripts or KO terms for all samples, without applying a fold change cutoff. The clustering

distance and clustering methods were "spearman" and "ward.D2", respectively. Clusters were identified with NbClust in R, focusing on the Gap statistic method with 500 bootstrap replicates. Clustering was used as a guide to identify major patterns and gene / annotation lists from the data. Venn diagrams and MA plots (log ratio versus mean expression) were generated in R using the limma and ggplot2 [44] packages.

### Functional annotation and gene set enrichment

Transcripts were functionally annotated by Diamond (BLASTX mode) comparisons against *Ectocarpus* Ec32 strain proteins. Gene ontology (GO) terms were appended to transcripts based on annotation data for the *Ectocarpus* genome (https://bioinformatics.psb.ugent.be/gdb/ectocarpusV2/). Predicted protein sequences derived from *S. latissima* reference transcripts were annotated against the InterPro [45] and KEGG [46] databases. Gene set enrichment analysis (GSEA) of GO terms was performed using clusterProfiler [47] in Bioconductor 3.8.

Transcripts with homology to genes from the male and female sex-determining regions (sdr) of *Ectocarpus* and their expression level in *S. latissima* were collated from functional annotation (Diamond BLASTX) and mapping (RSEM) data. Uniquely-expressed male and female transcripts were defined as those with average mapping $\geq 4$ transcripts per million (TPM) in the expressed sex and $\leq 1$ TPM in the non-expressed sex, identified from RSEM expression results.

## Results and discussion

### Transcriptome sequencing and expression profiles

High throughput sequencing resulted in 52.5 Gb of high quality paired-end read data (266.2 million male and 259.2 million female), with an average of 3.28 Gb (32.8 ± 1.04 million reads) per sample (S1 Table). Digital normalization reduced the dataset to 50.7 M reads for assembly with Velvet-Oases. After merging individual *k*-mer assemblies with transfuse 323,917 contigs remained with an N50 of 1,509 bp. After extracting and screening predicted open reading frames for Phaeophyceae-specific top hits, the final reference transcriptome consisted of 34,002 contigs with an N50 of 1,281 bp (S1 Table). Transcriptome completeness indicated that 64.1% of *Ectocarpus* protein coding genes were represented. According to BUSCO v2.0 analysis, the transcriptome was 74.9% complete (with 25.9% duplication level; S1 Table) and is therefore reasonably complete for a *de novo* transcriptome lacking representation from the sporophyte stage. This value rose to > 90% when fragmented or partial matches were considered. For comparison, running the same analysis with *Ectocarpus* strain Ec32 V2 proteins resulted in an estimate of 82.5% completeness and 9.1% duplication.

Differential expression (DE; over- or under-representation between sexes or between timepoints within a sex) was detected for 1,122 of the 11,916 transcripts analysed in edgeR (9.4%). A total of 3,429 KEGG orthologues were identified in the dataset, of which 521 (15.2%) were differentially expressed (a full list of KEGG-annotated DE genes is provided in S2 Table). Heatmap clustering of DE KO terms revealed several distinct patterns of gene regulation (Fig 2). The clustering indicated a primary divergence in expression between RL and WL, overriding differences due to gender. Further clustering of samples was primarily sex-dependent both during vegetative growth (RL) and gametogenesis. Transcript clusters revealed the presence of both female and male "constitutive" genes (Fig 2, clusters 1 and 8), with others representing transcripts predominantly up- or down-regulated during WL exposure (clusters 6 and 2, respectively), or showing transient or sex-dependent regulation (clusters 3–5,7).

Transcriptional profiles of female gametophytes are shown in Fig 3, as comparisons between vegetative (RL) growth conditions and the subsequent timecourse of gametogenesis

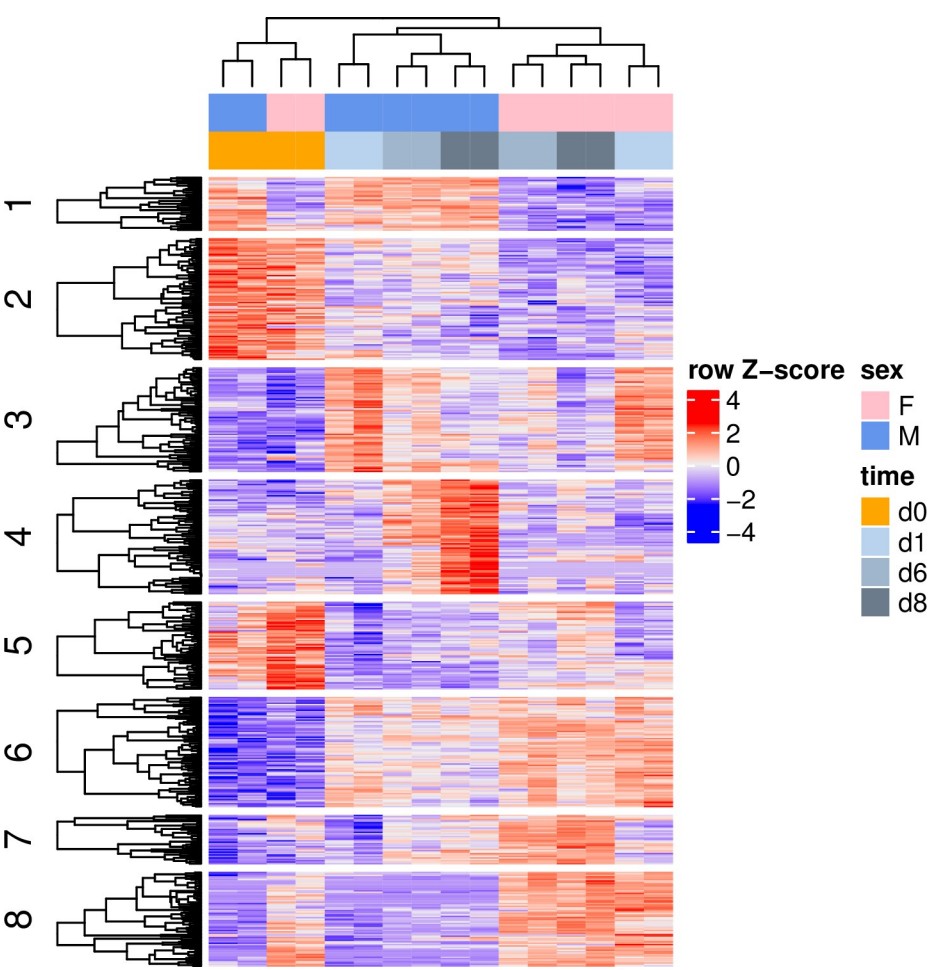

**Fig 2. Cluster analysis of differential gene expression.** Heatmap of 512 KEGG-annotated genes showing differential expression between gametophyte transcriptomes, either between sexes (F = female; M = male gametophytes) or timepoints (time = 0, 1, 6, 8 d). Expression values for each KEGG gene (row) are normalized across all samples (columns) by Z-score. Both column and row clustering were applied, and distinct gene clusters identified by the Gap statistic method are shown to illustrate the major expression patterns observed in the data.

after 1–8 days in WL. The majority of up-regulated KEGG-annotated genes appeared early (by 1 day) following exposure to WL (105 of the total 125). Of this total, 67 KEGG genes were uniquely up-regulated after 1 day, with only 20 genes additionally up-regulated after 6 and/or 8 days (Fig 3, upper Venn diagram). In male gametophytes, "early responsive" KEGG genes also dominated (Fig 4; 117 of 217), but the contribution of "late-responsive" genes was considerably higher than in females (100 additional KEGG genes after 6 and/or 8 days).

## Functional annotation

Gene set enrichment analysis (GSEA) of GO terms comparing the sexes showed that females were enriched for "ribosome", "translation" and related terms throughout the timecourse, suggesting that transcripts encoding ribosomal proteins are relatively more abundant in females during both vegetative growth and gametogenesis (S3 Table). After 1 and 6 days in WL, translational activity in females was also enriched ("translation initiation factor activity", "eukaryotic translation initiation factor 3 complex"; S3 Table). We also found GO:0055114 "oxidation-reduction process" overrepresented in female gametophytes (on day 6). This gene

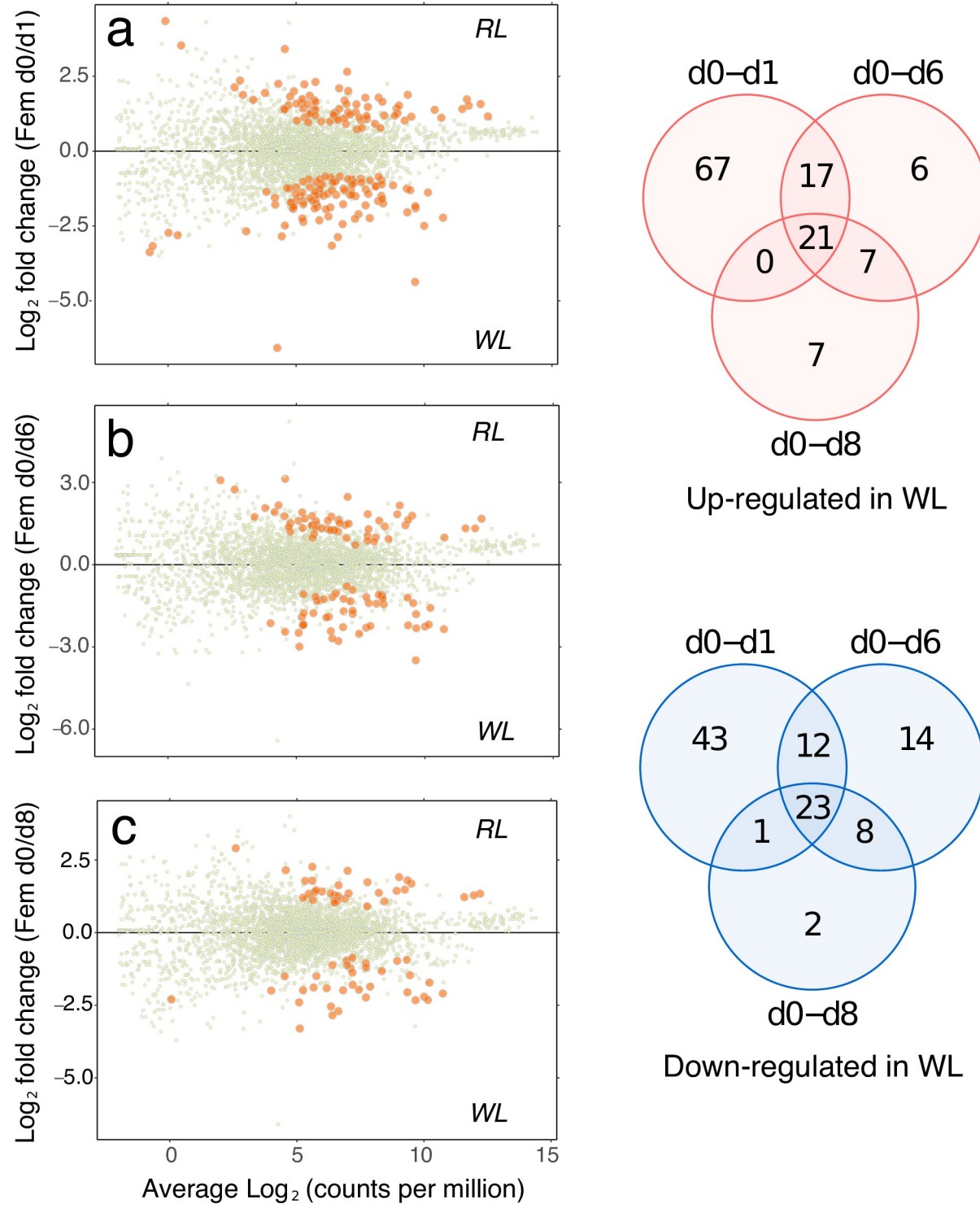

**Fig 3. Female gametophyte expression.** MA plots (log expression ratio vs. mean average expression) comparing female gametophyte gene expression (as KEGG-annotated genes) under vegetative growth (d0) with gametogenic conditions after a) 1 d, b) 6 d, and c) 8 d in WL. Each point on the plots represents a unique KEGG gene, with differentially expressed genes shown as larger orange points (edgeR, FDR < 0.05). The plots show expression on the x-axes as average $Log_2$ counts per million (CPM), and the ratio of RL/WL expression as $Log_2$(RL/WL) is shown on the y-axes. Venn diagrams summarise the expression changes across timepoints for genes up-regulated (upper Venn) and down-regulated (lower Venn) in gametogenic (WL) compared with vegetative (RL) conditions. A full list of KEGG gene annotations can be found in S2 Table.

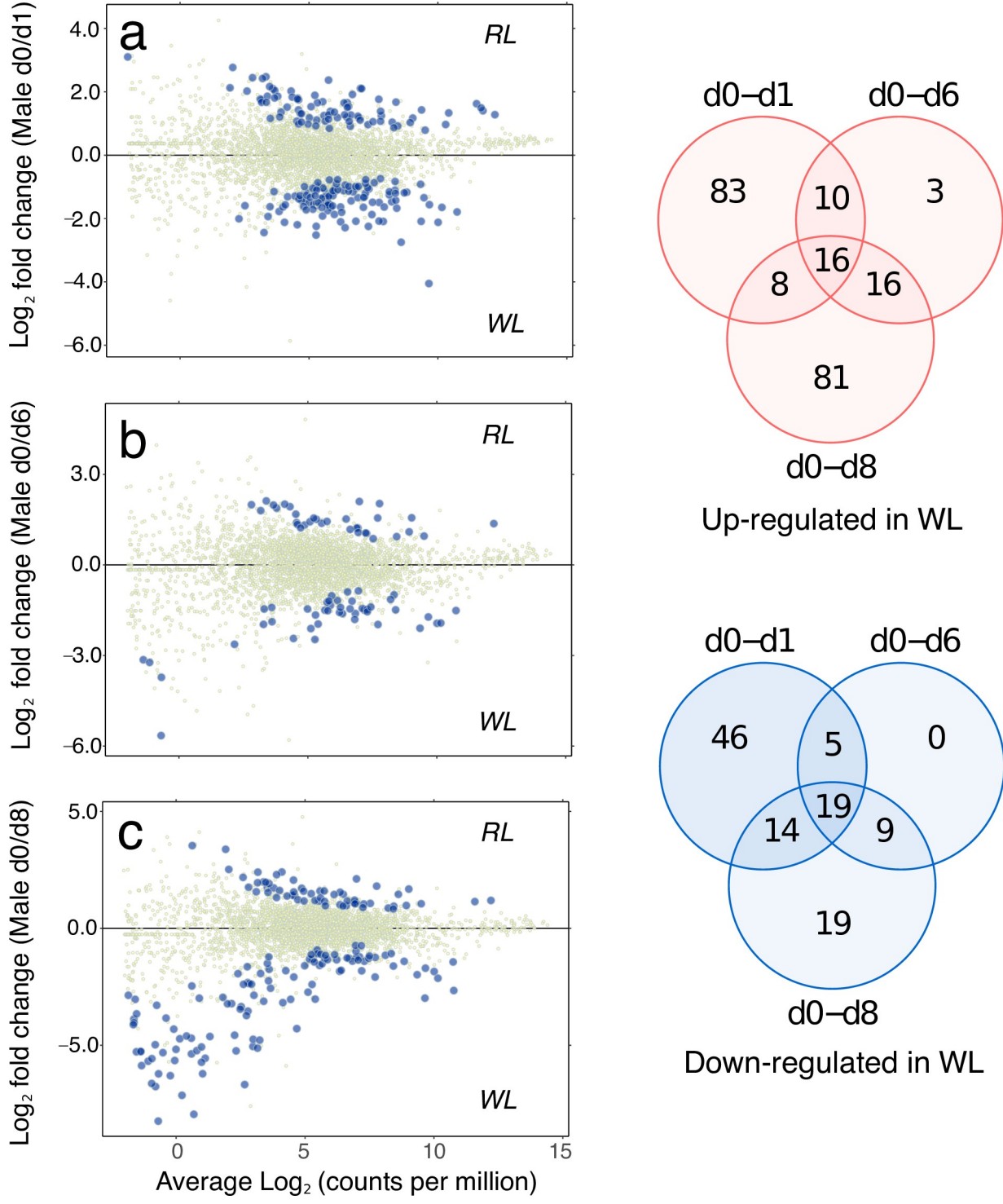

**Fig 4. Male gametophyte expression.** MA plots (log expression ratio vs. mean average expression) comparing male gametophyte gene expression (as KEGG-annotated genes) under vegetative growth (d0) with gametogenic conditions after a) 1 d, b) 6 d, and c) 8 d in WL. Each point on the plots represents a unique KEGG gene, with differentially expressed genes shown as larger orange points (edgeR, FDR < 0.05). The plots show expression on the x-axes as average $Log_2$ counts per million (CPM), and the ratio of RL/WL expression as $Log_2$(RL/WL) is shown on the y-axes. Venn diagrams summarise the expression changes across timepoints for genes up-regulated (upper Venn) and down-regulated (lower Venn) in gametogenic (WL) compared with vegetative (RL) conditions. A full list of KEGG gene annotations can be found in S2 Table.

set contains several uniquely or highly over-expressed genes in females, important among which seem to be group of genes related to wounding/pathogenesis. These include transcripts for a lipoxygenase, a respiratory burst oxidase, and a manganese SOD, all of which were female specific (with no detectable expression in male gametophytes).

Under vegetative growth conditions, male gametophytes showed enrichment of several terms, including "galactosylceramide sulfotransferase activity" and "golgi", containing several transcripts annotated as Galactose-3-O-sulfotransferases. This latter group of transcripts included 2 members uniquely expressed in males throughout gametogenesis (see "*Gender-biased or -specific expression in males*" below). Male vegetative gametophytes were also enriched in "ubiquitin-protein transferase activity" and "regulation of transcription, DNA-templated". For the latter, a range of transcription factors were represented, including heat shock factors (HSF), TFIIB, TFIIH, CCR4-Not (nuclear transcription from RNA polymerase II), sigma-70 (plastid transcription), as well as genes involved in chromatin remodelling (e.g., TAZ-type Zn finger protein, histone acetyltransferase). TALE-like homeodomain transcription factors were also represented, with homology to the *Ectocarpus ORO* life-cycle regulator [48, 49] as well as Ec-04_000450. Homologues for SAM, a second TALE transcription factor key to the regulation of life-cycle phase in *Ectocarpus*, which heterodimerizes with *ORO* [49] were not detected in our dataset. Following transfer to WL to induce gametogenesis, no over-represented GO terms were found in males (S3 Table).

## Gametogenesis involves regulation of key gender-independent "early response" genes

A majority of "early responsive" genes (i.e., significant expression changes after 1 day in WL) were common to both sexes, with 69 KEGG genes (or approximately two thirds) up-regulated in both males and females (S2 Table). These genes provide insights into general cellular, developmental and metabolic processes triggered during early gametogenesis.

From the expression changes in response to WL it can be inferred that the gametogenesis developmental program involves rapid activation of ribosome, transcription and translation related pathways. Furthermore, it was notable that several genes were involved in post-transcriptional or epigenetic regulation. An interesting example is SETD6 (K05302), a member of the SET domain family of protein lysine methyltransferases (PKMTs). The only differentially expressed member of six SET domain proteins identified in our dataset, SETD6 was immediately up-regulated under WL in both male and female gametophytes (Table 1). Reported targets of SETD6 methylation suggest a key role in gametogenesis. By methylating a histone H2A variant, SETD6 is directly implicated in the control of cellular differentiation in mouse embryonic stem cells [50]. SETD6 also functions in cell cycle regulation, via nonhistone methylation of the positive regulator of mitosis PLK1 (polo-like kinase 1), thereby exerting control over the rate of cell division [51].

In total, 11 KEGG genes belonging to the eukaryotic ribosome biogenesis pathway were up-regulated, including genes involved in rRNA 2′-O-methyation (NOP56, NOP58) and pseudouridylation (DKC1, NHP2) in the nucleolus, the co-translational acetylation of proteins (NAT10), the nuclear chaperone of 60S rRNA (midasin), as well as members of the UTP-B and MPP10 complexes associated with 90S pre-ribosomal RNA. A gene essential for 16S rRNA maturation and assembly of 30S ribosome subunits (ERAL1), as well as several genes involved in post-transcriptional modification (pseudouridylation) of tRNA and 23S rRNA were also up-regulated (DUS1, PUS7, rluB). Among other early responsive genes were several DEAD box RNA helicases, required for mitochondrial intron splicing (MSS116), and nuclear rRNA synthesis and processing (DBP3, DDX21), a Pumilio domain homology protein

**Table 1. Core "early responsive" KEGG genes in female and male gametophytes.**

| Gene | Description | KEGG gene | contigs | Log$_2$ fold-change compared to d0 (RL) | | | | | |
|------|-------------|-----------|---------|-------|-------|-------|-------|-------|-------|
| | | | | F: d1 | F: d6 | F: d8 | M: d1 | M: d6 | M: d8 |
| SETD6 | N-lysine methyltransferase SETD6 | K05302 | 1 | 7.0 | 6.2 | 4.6 | 3.0 | 2.4 | 2.1 |
| ERAL1 | GTPase | K03595 | 2 | 2.8 | 3.4 | 2.1 | 2.1 | 2.2 | 1.5 |
| DUS1 | tRNA-dihydrouridine synthase 1 | K05542 | 7 | 4.5 | 3.3 | 2.6 | 2.5 | 1.7 | 1.3 |
| PUS7 | tRNA pseudouridine13 synthase | K06176 | 8 | 3.5 | 2.3 | 1.8 | 3.1 | 2.5 | 1.8 |
| rluB | 23S rRNA pseudouridine2605 synthase | K06178 | 5 | 3.9 | 3.0 | 2.0 | 2.4 | 1.9 | 1.4 |
| NHP2 | H/ACA ribonucleoprotein complex subunit 2 | K11129 | 1 | 6.6 | 4.9 | 2.8 | 4.5 | 2.9 | 1.7 |
| DKC1 | H/ACA ribonucleoprotein complex subunit 4 | K11131 | 2 | 5.8 | 3.7 | 2.0 | 4.3 | 2.7 | 1.5 |
| NAT10 | N-acetyltransferase 10 | K14521 | 1 | 4.4 | 3.0 | 1.9 | 3.3 | 2.3 | 1.2 |
| UTP21 | U3 small nucleolar RNA-associated protein 21 | K14554 | 3 | 4.2 | 2.7 | 1.5 | 2.6 | 1.9 | 1.6 |
| UTP13 | U3 small nucleolar RNA-associated protein 13 | K14555 | 6 | 3.1 | 1.8 | 1.1 | 2.6 | 1.7 | 1.2 |
| UTP25 | U3 small nucleolar RNA-associated protein 25 | K14774 | 2 | 5.7 | 3.1 | 2.0 | 4.2 | 2.4 | 1.4 |
| IMP4 | U3 small nucleolar ribonucleoprotein protein IMP4 | K14561 | 2 | 7.3 | 4.5 | 2.9 | 3.8 | 2.4 | 1.7 |
| NOP56 | nucleolar protein 56 | K14564 | 5 | 4.9 | 3.6 | 1.9 | 4.2 | 3.1 | 2.0 |
| NOP58 | nucleolar protein 58 | K14565 | 4 | 3.6 | 2.6 | 1.4 | 3.4 | 2.6 | 1.6 |
| MDN1 | midasin | K14572 | 3 | 2.4 | 1.4 | 1.1 | 3.2 | 2.4 | 1.7 |
| PUF6 | pumilio homology domain family member 6 | K14844 | 1 | 5.1 | 2.8 | 1.6 | 3.8 | 2.2 | 1.6 |
| RRS1 | regulator of ribosome biosynthesis | K14852 | 1 | 4.0 | 2.7 | 1.6 | 3.9 | 2.7 | 1.5 |
| YTM1 | ribosome biogenesis protein YTM1 | K14863 | 2 | 5.0 | 2.9 | 1.9 | 6.2 | 3.1 | 2.8 |
| DBP3 | RNA helicase DBP3 | K14811 | 1 | 5.6 | 2.4 | 1.4 | 3.5 | 2.1 | 1.1 |
| DDX21 | RNA helicase DDX21 | K16911 | 5 | 3.9 | 2.5 | 1.2 | 2.6 | 1.4 | 0.7 |
| MSS116 | RNA helicase MSS116, mitochondrial | K17679 | 4 | 4.3 | 3.3 | 1.9 | 5.5 | 3.3 | 2.4 |
| EIF2S3 | translation initiation factor 2 subunit 3 | K03242 | 4 | 3.7 | 2.6 | 1.5 | 2.1 | 1.6 | 1.3 |
| EIF3F | translation initiation factor 3 subunit F | K03249 | 6 | 3.3 | 2.1 | 1.5 | 3.1 | 2.0 | 1.5 |
| EIF3B | translation initiation factor 3 subunit B | K03253 | 4 | 4.2 | 2.8 | 1.9 | 2.9 | 2.0 | 1.3 |
| EIF4A | translation initiation factor 4A | K03257 | 5 | 3.7 | 2.9 | 1.7 | 3.4 | 2.1 | 1.5 |
| NRT | MFS transporter, NNP family, nitrate/nitrite transporter | K02575 | 7 | 4.4 | 3.5 | 3.2 | 4.1 | 3.2 | 3.0 |
| gdhA | glutamate dehydrogenase (NADP+) | K00262 | 9 | 1.9 | 2.3 | 2.3 | 1.6 | 1.7 | 1.7 |
| GLT1 | glutamate synthase (NADH) | K00264 | 2 | 1.6 | 1.2 | 1.1 | 1.4 | 1.1 | 0.9 |
| gltD | glutamate synthase (NADPH) small chain | K00266 | 2 | 1.5 | 1.7 | 1.4 | 1.2 | 0.9 | 0.9 |

Core transcription- and translation-related (above the line) and nitrogen metabolism (below the line) "early responsive" KEGG genes upregulated in male and female gametophytes of *S. latissima* within 24 h of exposure to gametogenic conditions. Average Log$_2$ fold-change values compared with vegetative conditions (RL) are given for females (F) and males (M) after 1, 6, and 8 days exposure to WL. The corresponding number of annotated *S. latissima* contigs representing each KEGG gene are shown. See S2 Table for a complete list of "early responsive" genes.

implicated in translational control of mRNA, as well as several translation initiation factors (Table 1 and S2 Table). Ribosome biogenesis and protein synthesis gene networks play key roles in controlling germline stem cell differentiation in animals [52], and mutations affecting rRNA processing also affect plant gametogenesis and embryogenesis [53, 54]. Although our data are limited to observations of transcript levels, the evidence for involvement of similar pathways across independent multicellular lineages appears compelling. The central role for transcriptional and translational control in gametogenesis suggests that proteomic comparisons would complement RNAseq-based approaches. Another promising future direction might be mutant screens in the model brown algal system (such as *Ectocarpus*) to analyse phenotypes impaired in gamete formation.

The rapid regulation/reorganization of the transcriptional and translational machinery was accompanied by metabolic adjustment. Notably, several key genes in nitrogen metabolism were upregulated (Table 1). These included strong up-regulation (ca. 20-fold) of a nitrate/ nitrite transporter (*NRT*), suggesting an increased requirement for inorganic N acquisition for gametogenesis, reminiscent of the response of diatoms under N-deprivation [55]. Although N and P levels in the medium were lower after transfer to WL gametogenic conditions, and we cannot exclude an effect on uptake rates, the medium was still nutrient replete. N-acquisition was accompanied by upregulation of ammonium assimilation genes contributing to cellular pools of L-glutamate including both chloroplastic *Fd-GOGAT* (*GLT1*) and cytosolic *NADPH-- GOGAT* (*gltD*), and possibly *GDH* (but see [56, 57]). Biosynthesis of several amino-acids depends on available glutamate (e.g., alanine, aspartate and glutamate metabolism, arginine metabolism) and these pathways also showed evidence of upregulation during gametogenesis.

Pyruvate is both the product of glycolysis and starting metabolite for gluconeogenesis, and acts as a key metabolite in interacting energy and carbon metabolism pathways. In the penulti- mate and final steps of glycolysis to generate pyruvate, enolase (*ENO*) and pyruvate kinase (*PK*) were upregulated, as was pyruvate, phosphate dikinase (*PPDK*) which reverses the action of *PK* to generate phosphoenolpyruvate (S2 Table). Pyruvate and glutamate are also produced by transamination of alanine by alanine transaminase (*ALT*), which was strongly upregulated (S2 Table). Pyruvate is the precursor for acetyl CoA entering the TCA (citric acid cycle), which was moderately upregulated together with pyruvate and diydrolipoamide dehydrogenases (*PDHA*, *DLAT* and *DLD*).

We observed no clear pathway-level responses among down-regulated "early responsive" genes, although we noted a transitory down-regulation of light-harvesting (*LHCA1* and *4*) and photosynthesis proteins (*psbQ*, *psbU*). This was, however, followed by subsequent up-regula- tion at days 6–8, and can likely be attributed to increased excitation pressure on PS2 caused by lower temperature and higher light after the transition from RL to WL [58].

## Gender-biased or -specific expression in females

The identification of "core" genes constitutively over- (or uniquely) expressed in the two gen- ders can provide insight into sex-specific genetic networks and developmental programs. In female gametophytes, 15 KEGG genes were statistically assigned to the core set (Fig 5A, 5C– 5F). These included several genes with no detectable expression in male gametophytes that thus may provide insight into the control of female gametogenesis (Table 2). They include a transcript with repeated RCC1 (regulator of chromosome condensation) motifs and annotated as KEGG orthologue HERC3 (K10614, an E3 ubiquitin-protein ligase). We also identified a female-specific Ran GTPase (SL_90811), for which RCC1 is the guanine nucleotide exchange factor. Together they are thought to play an important role in nucleocytoplasmic transport and mitotic regulation [59]. Another potential HERC3 interacting protein is KCTD9 (K21919), which in humans interacts with the E3 ubiquitin ligase complex to mediate the ubi- quination of target proteins for degradation, and which was expressed (at low levels) in WL only in female gametophytes. The pre-mRNA splicing factor SYF1 (an LSM3 isoform and component of the spliceosome) was also uniquely expressed in female gametophytes (Fig 5A, 5C–5F). Taken together, these results suggest that several factors are involved in the control of female-specific gene expression and the female gametogenic developmental program in *S. latissima*.

Two contigs annotated as *myo*-inositol-1-phosphate synthase (INO1; K01858) were present in our dataset, homologous to *Ectocarpus* Ec-26_005440 (with 95 and 74% amino acid identity, respectively). However, while the most homologous copy was equally expressed in male and

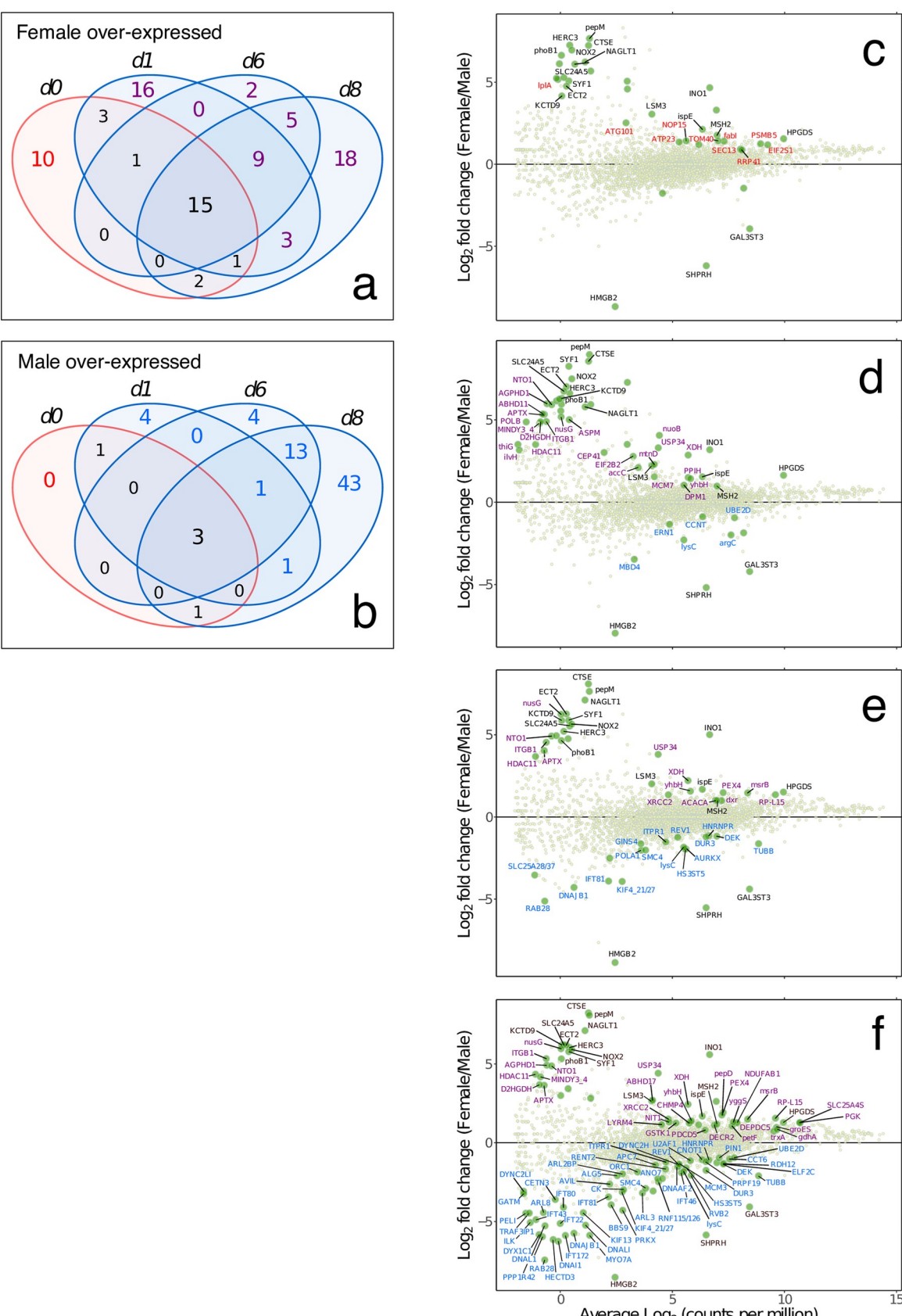

**Fig 5. Comparative expression in males and females during gametogenesis.** Venn diagrams showing numbers of differentially expressed genes (KEGG-annotated genes) upregulated at each experimental timepoint in females a) and males b). Differentially expressed genes under vegetative conditions only are highlighted in red, "core" genes overexpressed in all conditions in one of the sexes are highlighted in bold, while WL-responsive genes are highlighted in purple (females) or blue (males). MA plots (log expression ratio vs. mean average expression) of female vs. male gametophyte expression of KEGG-annotated genes in vegetative growth conditions c) 0 days, and after d) 1 day, e) 6 days and f) 8 days of culture under WL gametogenic conditions. Each point represents a unique KEGG gene, with differentially expressed shown as larger green points (edgeR; FDR < 0.05) and gene labels are colour-coded according to the Venn diagrams to the left (vegetative = red; "core" = black; WL-responsive = purple or blue for females and males, respectively). The plots show expression on the x-axes as average $Log_2$ counts per million (CPM), and the ratio of female/male expression as $Log_2$(F/M) is shown on the y-axes. KEGG gene information: **1) Female vegetative (RL)**: *lplA*: lipoate-protein ligase; *ATG101*: autophagy-related protein 101; *ATP23*: mitochondrial inner membrane protease; *NOP15*: nucleolar protein 15; *TOM40*: mitochondrial import receptor subunit; *SEC13*: protein transport protein; *fabI*: enoyl-[acyl-carrier protein] reductase I; *RRP41*: exosome complex component; *PSMB5*: 20S proteasome subunit beta 5; *EIF2S1*: translation initiation factor 2 subunit 1. **2) Female "core" up-regulated**: *HERC3*: E3 ubiquitin-protein ligase; *phoB1*: two-component system, OmpR family, alkaline phosphatase synthesis response regulator; *pepM*: phosphoenolpyruvate phosphomutase; *CTSE*: cathepsin E; *NOX2*: NADPH oxidase 2; *NAGLT1*: MFS transporter, FHS family, Na+ dependent glucose transporter 1; *SLC24A5*: solute carrier family 24 (sodium/potassium/calcium exchanger), member 5; *SYF1*: pre-mRNA-splicing factor; *ECT2*: protein ECT2; *KCTD9*: BTB/POZ domain-containing protein; *LSM3*: U6 snRNA-associated Sm-like protein; *INO1*: myo-inositol-1-phosphate synthase; *ispE*: 4-diphosphocytidyl-2-C-methyl-D-erythritol kinase; *MSH2*: DNA mismatch repair protein; *HPGDS*: prostaglandin-H2 D-isomerase / glutathione transferase. **3) Male "core" up-regulated**: *HMGB2*: high mobility group protein B2; *SHPRH*: E3 ubiquitin-protein ligase; *GAL3ST3*: galactose-3-O-sulfotransferase 3. **4) Female WL-responsive**: *ABHD11*: abhydrolase domain-containing protein 11; *ABHD17*: abhydrolase domain-containing protein 17; *ACACA*: acetyl-CoA carboxylase / biotin carboxylase 1; *accC*: acetyl-CoA carboxylase, biotin carboxylase subunit; *AGPHD1*: hydroxylysine kinase; *APTX*: aprataxin; *ASPM*: abnormal spindle-like microcephaly-associated protein; *CEP41*: centrosomal protein; *CHMP4*: charged multivesicular body protein 4; *D2HGDH*: D-2-hydroxyglutarate dehydrogenase; *DECR2*: peroxisomal 2,4-dienoyl-CoA reductase; *DEPDC5*: DEP domain-containing protein 5; *DPM1*: dolichol-phosphate mannosyltransferase; *dxr*: 1-deoxy-D-xylulose-5-phosphate reductoisomerase; *EIF2B2*: translation initiation factor eIF-2B subunit beta; *gdhA*: glutamate dehydrogenase (NADP+); *groES*: chaperonin GroES; *GSTK1*: glutathione S-transferase kappa 1; *HDAC11*: histone deacetylase 11; *ilvH*: acetolactate synthase I/III small subunit; *ITGB1*: integrin beta 1; *LYRM4*: LYR motif-containing protein 4; *MCM7*: DNA replication licensing factor; *MINDY3_4*: ubiquitin carboxyl-terminal hydrolase; *msrB*: peptide-methionine (R)-S-oxide reductase; *mtnD*: 1,2-dihydroxy-3-keto-5-methylthiopentene dioxygenase; *NDUFAB1*: NADH dehydrogenase (ubiquinone) 1 alpha/beta subcomplex 1; *NIT1*: deaminated glutathione amidase; *NTO1*: NuA3 HAT complex component; *nuoB*: NADH-quinone oxidoreductase subunit B; *nusG*: transcriptional antiterminator; *PDCD5*: programmed cell death protein 5; *pepD*: dipeptidase D; *petF*: ferredoxin; *PEX4*: peroxin-4; *PGK*: phosphoglycerate kinase; *POLB*: DNA polymerase beta; *PPIH*: peptidyl-prolyl isomerase H (cyclophilin H); *RP-L15*: large subunit ribosomal protein L15; *SLC25A4S*: solute carrier family 25 (mitochondrial adenine nucleotide translocator); *thiG*: thiazole synthase; *trxA*: thioredoxin 1; *USP34*: ubiquitin carboxyl-terminal hydrolase 34; *XDH*: xanthine dehydrogenase/oxidase; *XRCC2*: DNA-repair protein; *yggS*: PLP dependent protein; *yhbH*: putative sigma-54 modulation protein. **5) Male WL-responsive**: *ALG5*: dolichyl-phosphate beta-glucosyltransferase; *ANO7*: anoctamin-7; *APC7*: anaphase-promoting complex subunit 7; *argC*: N-acetyl-gamma-glutamyl-phosphate reductase; *ARL2BP*: ADP-ribosylation factor-like protein 2-binding protein; *ARL3*: ADP-ribosylation factor-like protein 3; *ARL8*: ADP-ribosylation factor-like protein 8; *AURKX*: aurora kinase; *AVIL*: advillin; *BBS9*: Bardet-Biedl syndrome 9 protein; *CCNT*: cyclin T; *CCT6*: T-complex protein 1 subunit zeta; *CETN3*: centrin-3; *CK*: creatine kinase; *CNOT1*: CCR4-NOT transcription complex subunit 1; *DEK*: protein DEK; *DNAAF2*: dynein assembly factor 2, axonemal; *DNAI1*: dynein intermediate chain 1, axonemal; *DNAJB1*: DnaJ homolog subfamily B member 1; *DNAL1*: dynein light chain 1, axonemal; *DNALI*: dynein light intermediate chain, axonemal; *DUR3*: urea-proton symporter; *DYNC2H*: dynein heavy chain 2, cytosolic; *DYNC2LI*: dynein light intermediate chain 2, cytosolic; *DYX1C1*: dyslexia susceptibility 1 candidate gene 1 protein; *ELF2C*: eukaryotic translation initiation factor 2C; *ERN1*: serine/threonine-protein kinase/ endoribonuclease IRE1; *GATM*: glycine amidinotransferase; *GINS4*: GINS complex subunit 4; *HECTD3*: E3 ubiquitin-protein ligase; *HNRNPR*: heterogeneous nuclear ribonucleoprotein R; *HS3ST5*: [heparan sulfate]-glucosamine 3-sulfotransferase 5; *IFT22*: intraflagellar transport protein 22; *IFT43*: intraflagellar transport protein 43; *IFT46*: intraflagellar transport protein 46; *IFT80*: intraflagellar transport protein 80; *IFT81*: intraflagellar transport protein 81; *IFT172*: intraflagellar transport protein 172; *ILK*: integrin-linked kinase; *ITPR1*: inositol 1,4,5-triphosphate receptor type 1; *KIF4_21_27*: kinesin family member; *KIF13*: kinesin family member 13; *lysC*: aspartate kinase; *MBD4*: methyl-CpG-binding domain protein 4; *MCM3*: DNA replication licensing factor; *MYO7A*: myosin VIIa; *ORC1*: origin recognition complex subunit 1; *PELI*: pellino; *PIN1*: peptidyl-prolyl cis-trans isomerase NIMA-interacting 1; *POLA1*: DNA polymerase alpha subunit A; *PPP1R42*: protein phosphatase 1 regulatory subunit 42; *PRKX*: protein kinase X; *PRPF19*: pre-mRNA-processing factor 19; *RAB28*: Ras-related protein; *RDH12*: retinol dehydrogenase 12; *RENT2*: regulator of nonsense transcripts 2; *REV1*: DNA repair protein; *RNF115_126*: E3 ubiquitin-protein ligase; *RVB2*: RuvB-like protein 2; *SLC25A28_37*: solute carrier family 25 (mitochondrial iron transporter); *SMC4*: structural maintenance of chromosome 4; *TRAF3IP1*: TRAF3-interacting protein 1; *TUBB*: tubulin beta; *U2AF1*: splicing factor U2AF 35 kDa subunit; *UBE2D*: ubiquitin-conjugating enzyme E2 D.

female gametophytes, the other had higher overall expression but was undetectable in males (Table 2). Amino acid translations indicated that both encode functional full-length INO1 gene copies, based on proteinalignments and InterPro annotations. Furthermore, phylogenetic analysis of available stramenopile INO1 proteins placed both copies in the brown algal clade (S1 File). Given the level of sequence variation between *S. latissima* INO1 copies, it should in future be possible to verify these data from genomic DNA, and study their expression in more

**Table 2. Over-expressed genes in female gametophytes.**

| KEGG KO | KEGG gene | Description | Accession | Identity (%) | logFC$_{max}$ (F:M) | Time (d) |
|---------|-----------|-------------|-----------|--------------|---------------------|----------|
| K10614 | HERC3 | E3 ubiquitin-protein ligase | CBN79063.1 | 30.9 | 7.26 | 0, 1, 6, 8 |
| K21919 | KCTD9 | BTB/POZ domain-containing protein | CBJ32546.1 | 43.6 | 6.31 | 0, 1, 6, 8 |
| K07936[a] | RAN | GTP-binding nuclear protein | CBJ27657.1 | 81.5 | 8.95 | 0, 1, 6, 8 |
| K12867 | SYF1 | pre-mRNA-splicing factor | SJ08841[b] | 35.0 | 8.25 | 0, 1, 6, 8 |
| K01858[c] | INO1 | myo-inositol-1-phosphate synthase | CBN77493.1 | 74.2 | 12.10 | 0, 1, 6, 8 |
| *K01858*[d] | *INO1* | *myo-inositol-1-phosphate synthase* | *CBN77493.1* | *91.2* | *-1.36* | *nsig* |
| K21421 | NOX2 | NADPH oxidase 2 | CBJ31029.1 | 31.3 | 7.49 | 0, 1, 6, 8 |
| K04564[e] | SOD2 | superoxide dismutase, Fe-Mn family | CBN79353.1 | 41.1 | 9.42 | 0, 1, 6, 8 |
| K04097[f] | HPGDS | prostaglandin-H2 D-isomerase / glutathione transferase | SJ22284 | 39.3 | 9.85 | 0, 1, 6, 8 |

KEGG genes and contigs of *S. latissima* over-expressed in female versus male gametophytes. The corresponding accession numbers of *Ectocarpus* Ec32 or *S. japonica* and percentage protein identity are shown. Also indicated are the maximum fold-change (Log$_2$ FCmax) observed and the sampling point(s) in the timecourse for which differentiation expression was detected (BH adjusted P ≤ 0.05). For INO1, data are shown for female specific and non-biased isoforms.

[a]SL_90811

[b]*Saccharina japonica* (Ye *et al.* 2015). No homologue found in *Ectocarpus*

[c]SL_25960; Female specific

[d]SL_25748; non sex-biased

[e]SL_77691

[f]Nine of 12 contigs over- or uniquely-expressed in females. Results shown are for SL_85204

detail using qPCR. INO1 is a key enzyme catalysing the first (and rate limiting) step in the production of inositol-containing compounds from D-glucose 6-phosphate, thereby playing a central role in phospholipid biosynthesis and phosphatidylinositol signalling [60] and various other processes such as phosphate storage in plants, tolerance to abiotic stress, and morphogenesis [61–64]. We were, however, unable to identify additional gene expression changes in support of specific downstream pathways involving inositol.

Reactive oxygen species (ROS) are emerging as important signalling molecules that regulate a wide range of physiological and developmental processes, in addition to their potential cytotoxic effects. An important role of ROS signalling during female gametogenesis has been implicated in both mammalian [65] and plant oogenesis [66]. In *S. latissima* we found two genes with female-specific expression that suggest ROS signalling may also be important in brown algal oogenesis (Table 2); a membrane localized superoxide generating NADPH oxidase (NOX; K21421) was constitutively expressed in female gametophytes, and an Fe-Mn SOD (SOD2; K04564), which showed a 7.5-fold increase in expression after transfer from RL to WL. SOD has been implicated in the regulation of ROS levels in plant megagametogenesis [67]. These observations suggest the working hypothesis that signalling via an oxidative burst may be involved in *S. latissima* oogenesis.

Perhaps related to putative ROS signalling discussed above, we identified two KEGG genes involved in the prostaglandin (PG) synthesis pathway, prostaglandin-E synthase 2 (PTGES2) and prostaglandin-H2 D-isomerase (HPGDS). Significantly, 9 out of 12 contigs for HPGDS were strongly and constitutively female-biased (Fig 5C–5F; Table 2). PGs are eicosanoids derived enzymatically from 20-carbon polyunsaturated fatty acids (PUFAs), universally present in animals where they have diverse hormonal effects. PG have been detected in brown algae, first identified from Laminariales in the context of stress responses to heavy metal exposure [68]. More recently, gene sequences for PTGES2 and HPGDS were confirmed from diatoms (unicellular heterokont algae) [69]. However, we believe this is the first report of sex-biased expression of HPGDS in algae, which clearly deserves further investigation.

## Gender-biased or -specific expression in males

Only three KEGG genes were detected as constitutively overexpressed in male relative to female gametophytes (Fig 5B, 5C–5F). Prominent among these was a high mobility group (HMGB2) protein, represented by a single contig uniquely expressed in males. HMG proteins are transcription factors involved in sex-determination in both animals and fungi [70, 71]. The *S. latissima* sequence was homologous to *Ectocarpus* Ec-13_001750, a locus first identified from the male sex-determining region (SDR) in the genome of *Ectocarpus* [12] and implicated as the potential male-determining factor in these brown algae [13].

Unsurprisingly, an array of genes with mainly late-onset upregulation (days 6 and 8) confirmed the important role of cell proliferation in male gametogenesis (production of large numbers of sperm relative to eggs) at this stage in the time course, and revealed several parallels at the molecular level with plant and animal gametogenesis (Fig 5B, 5C–5F; S2 Table). A total of 4 E3 ubiquitin-protein ligases were over-expressed in males. One of these, SHPRH (K15710) was constitutively male-specific (i.e., TPM < 1 in females; S2 Table; Fig 5C–5F). SHPRH is thought to promote error-free replication in proliferating cells [72], including during mammalian spermatogenesis [73]. The constitutively male-biased expression of this gene may point to a key upstream role in male development. Two of the three other male-biased E3 ubiquitin-protein ligases were members of the RING-type (K110964, K11982), and all have putative roles in cell cycle progression or membrane trafficking. The sex-biased expression of various distinct E3 ubiquitin-protein ligases (with both male- and female-biased members identified) highlights their potential importance as specific and selective regulatory components of sexual development in brown algae, in common with other systems [74]. Several known markers for cell proliferation were male-biased; aurora kinase, a GINS complex subunit, Centrin 3 and MCM3, as well as protein DEK (variously involved in regulation of chromatin structure, epigenetic modification, and transcription) [75]. These data indicate that several common elements of the cell proliferation molecular machinery in multicellular lineages are invoked during spermatogenesis in *S. latissima*.

The sulfotransferase (ST) GAL3ST3 (galactose-3-O-sulfotransferase 3) was represented by 5 contigs with homology to known brown algal proteins. The most highly expressed member was male-specific in *S. latissima*, likely reflecting differences in the regulation of cell wall structure between male and female gametophytes. It was shown more than 30 years ago in *S. latissima* that the antheridial cap, which ruptures to allow the explosive release of sperm in response to female pheromone release, contains sulphated polysaccharides [76] and is therefore a potential site of action for these gene products in males. It was also shown by [76] that antheridial discharge requires Na+, with a rapidity suggesting the possible involvement of a membrane action potential. While not supported statistically, expression of a voltage-gated Na + channel (K04834; NAV1.2) was detected only in males after 8 days, and is therefore potentially interesting in the context of sperm release.

## Expression of flagella-associated genes (mainly) in males

Prominent among male-biased genes expressed after 8 days in WL (Fig 5F) were a number contributing to flagellar development in the antheridia. Several KEGG genes for intraflagellar transport (IFT) and dynein assembly proteins showed low but significantly up-regulated expression after day 8 in WL (Fig 5E and 5F; Table 3), while expression in females was generally very low or not detected, and invariant over the timecourse. The IFT system consists of protein complexes responsible for flagellar assembly and maintenance [77], whose upregulation would be expected during flagellar biogenesis. Several other flagella-related genes (axonemal dyneins, ciliogenesis-related proteins) were upregulated in male gametophytes after day 8

**Table 3. Flagella-related genes over-expressed in male gametophytes.**

| KEGG KO | KEGG gene | Description | *Ectocarpus* Acc | [a]Identity (%) | Log$_2$ FC$_{max}$ (M:F) | Time (d) |
|---------|-----------|-------------|------------------|-----------------|--------------------------|----------|
| K07935 | IFT22, RABL5 | intraflagellar transport protein 22 | CBJ29852.1 | 76.2 | 5.12 | 8 |
| K10409 | DNAI1 | dynein intermediate chain 1, axonemal | CBJ32625.1 | 99.0 | 6.25 | 8 |
| K10410 | DNALI | dynein light intermediate chain, axonemal | CBN75111.1 | 94.9 | 5.23 | 8 |
| K10411 | DNAL1 | dynein light chain 1, axonemal | CBJ48309.1 | 88.4 | 5.95 | 8 |
| K19398 | BBS9 | Bardet-Biedl syndrome 9 protein | CBN75355.1 | 82.9 | 3.93 | 8 |
| K19584 | PRKX | protein kinase X [EC:2.7.11.11] | CBJ25898.1 | 92.2 | 4.28 | 8 |
| K19675 | IFT43 | intraflagellar transport protein 43 | CBN74139.1 | 79.4 | 4.87 | 8 |
| K19676 | IFT172 | intraflagellar transport protein 172 | CBN75458.1 | 92.6 | 5.87 | 8 |
| K19677 | IFT81 | intraflagellar transport protein 81 | CBJ33391.1 | 76.8 | 3.43 | 6, 8 |
| K19678 | IFT80 | intraflagellar transport protein 80 | CBN79966.1 | 85.6 | 4.09 | 8 |
| K19680 | TRAF3IP1, IFT54 | TRAF3-interacting protein 1 | CBJ27365.1 | 82.4 | 4.46 | 8 |
| K19682 | IFT46 | intraflagellar transport protein 46 | CBJ28183.1 | 80.2 | 1.89 | 8 |
| K19683 | TTC30, DYF1 | tetratricopeptide repeat protein 30 | CBJ30057.1 | 88.4 | 1.92 | 8 |
| K19751 | DNAAF2, KTU, PF13 | dynein assembly factor 2, axonemal | CBJ29402.1 | 75.1 | 1.52 | 8 |
| K19758 | DYX1C1, DNAAF4 | dyslexia susceptibility 1 candidate gene 1 protein | CBJ30590.1 | 49.2 | 5.82 | 8 |
| K22866 | TCTEX1D2 | tctex1 domain-containing protein 2 | CBN78115.1 | 91.2 | 2.39 | 8 |
| **Contig** | | | | | | |
| SL_17525 | na | flagellar associated protein putative | CBJ48496.1 | 93.5 | 3.11 | 8 |
| SL_34693 | na | flagellar associated protein putative | CBN76272.1 | 57.6 | 4.56 | 8 |

Flagella-related KEGG genes and contigs of *S. latissima* over-expressed in male versus female gametophytes. The corresponding accession number of *Ectocarpus* Ec32 and percentage protein identity are shown. Also indicated are the maximum fold-change (Log$_2$ FC$_{max}$) observed and the sampling point(s) in the timecourse for which differentiation expression was detected (BH adjusted $P \leq 0.05$).

[a]Where more than 1 contig is involved, the highest-scoring is reported.

in WL (Table 3), suggesting that antheridial maturation and sperm development were ongoing processes at this time. We comprehensively analysed the expression of flagella-related genes in both female and male gametophyte transcriptomes, based on annotated proteins from the genome of *Ectocarpus* Ec32 [4] and previous proteomic analysis of brown algal flagella [78]. Of the 93 flagella-related genes identified in the *Ectocarpus* genome [4], 91 homologues were identified in our *S. latissima* transcriptome (S4 Table). Expression of flagellar-related genes was highly male-biased, with 45 genes expressed only in males (i.e., female read mapping $\leq 1$ TPM, defined as the threshold for expression). However, a further 45 genes were expressed by both sexes, and female-specific contigs were found for 3 genes (S4 Table). In future, it will be important to obtain male and female gametophyte genomes for deep RNAseq mapping to explore sex-specific transcriptional variation and/or post-transcriptional variation (e.g., alternative splice variants).

While flagella are associated chiefly with motile reproductive cells of brown algae, such as sperm, and sexual reproduction in Laminariales is oogamous, it has been known for over 30 years that eggs of Laminariales retain the ability to produce flagella [79]. These lack a function in motility, however, while possessing distinct features (an absence of mastigonemes and divergent basal body structure). Rather than purely vestigial organelles remaining after the evolutionary transition from anisogamy to oogamy, their persistence post-fertilization suggests an important role in anchoring the developing zygote to the oogonium, potentially influencing zygote polarity in early sporophyte development [80]. It remains for future studies to determine the detailed timing and extent of possible female flagella-related gene expression, perhaps at later stages of egg development.

## Conclusions

This study provides the first overview of expression differences during gametogenesis in kelps in the context of sex-bias and developmental trajectory. Although preliminary, the results provided an unexpectedly rich picture of extensive changes in transcriptional profiles triggered by gametogenic conditions in *S. latissima*. Transcriptome profiling revealed that a common immediate response of both sexes to gametogenic conditions overrides sex-specific transcriptional changes, with specific gene annotations suggesting an important role for post-transcriptional and epigenetic regulation of ribosome biogenesis, cell proliferation and differentiation. In this phase, key genes involved in nutrient (N) assimilation and control of intersecting energy and carbon metabolic pathways also showed evidence of regulatory changes.

Consistent with the idea that female development may be the default pathway in brown algal UV sexual systems [12], a number of female-biased or unique genes were identified with roles in gene expression regulation, while males uniquely expressed the putative sex-determining HMG factor protein, hypothesised to repress female development. Male-biased gene expression appears largely to function in coordinating cell proliferation during sperm production, and for sperm flagella biosynthesis. The results also highlight the potential roles of E3 ubiquitin-protein ligases in sex-specific gametogenic pathways, and of ROS signalling in female gametogenesis, both with either ancient or convergent parallels in evolutionarily divergent multicellular lineages.

These and other observations (e.g., on timing and sex-specificity of flagella development), whilst preliminary, suggest a fascinating array of genes, pathways and processes that may be targeted in future functional and comparative studies of brown algal gametogenesis.

## Supporting information

**S1 File. Alignment and phylogenetic analysis of Myo-inositol-1-phosphate synthase (INO1) predicted proteins from stramenopiles.** A. Alignment in nexus format of 38 INO1 amino acid sequences obtained after BLASTP analysis (NCBI nr) of 2 INO1 predicted proteins identified in the *S. latissima* transcriptome (SL_25748.1_384_1991_- and SL_25960.2_333_2118_-). The alignment was performed using Muscle[1] and curated with Gblocks[2] and low stringency parameters to remove gaps and poorly-aligned regions, resulting in 486 amino acid characters. B. ML phylogenetic tree of stramenopile INO1 protein sequences. The tree was built using PhyML[3] (LG model; aLRT branch support; model-given amino-acid frequencies; optimized across-site rate variation; best of NNI and SPR tree search). *S. latissima* INO1 contigs are well-supported within the Phaeophyceae (brown algal) clade. The non sex-biased copy SL_25748.1 is sister to the *S. japonica* INO1. The female-biased copy SL_25960.2 is more divergent. It should be noted that following Lipinska et al. (2017)[4], the *S. japonica* genome sequence derives from a male gametophyte strain.
(DOCX)

**S1 Table. Read data, assembly and annotation statistics.** A) Read data used in the study. B) Assembly statistics for initial assembly (Velvet-Oases) and final reference after merging with transfuse. C) Basic annotation statistics for reference *S. latissima* transcriptome.
(XLSX)

**S2 Table. KEGG-annotated gene lists for differentially expressed transcripts in *Saccharina latissima*.** Annotations and gene descriptions are given, together with average expression values (TPM; transcripts per million), expression ratios (Log$_2$ fold-change) and statistical support for comparisons (BH-adjusted *P*-values). RL = red light; vegetative growth conditions. WL = white light; gametogenic conditions.
(XLSX)

**S3 Table. Results of gene set enrichment analysis (GSEA).**
(XLSX)

**S4 Table. Flagella-related proteins in *S. latissima*.** Check list of flagella-related proteins annotated from the *Ectocarpus* Ec32 genome (from [4], Suppl. Table 42), and the corresponding number of contigs identified in developing gametophytes of *Saccharina latissima*. Expression patterns were assessed from read mapping, where $\geq 2$ TPM (transcripts per million) is considered as expressed. MF = contigs expressed in both males and females, M = only expressed in males, F = only expressed in females. Potential contamination was assessed from phylogenetic analysis of contigs and all Blastx hits (Stramenopile protein database; $E \leq e^{-10}$). Amino acid alignments were built using Muscle (Edgar 2004), and trees were generated using PhyML (Guindon et al. 2010), with LG model and aLRT support values. Contigs with non-sister relationships to Phaeophyceae were considered to derive from contamination. Missing data represent cases where contigs were fragmented and alignment was not possible or unreliable.
(XLSX)

## Acknowledgments

We thank A. Wagner for maintaining the algal material and for help with laboratory facilities.

## Author Contributions

**Conceptualization:** Gareth A. Pearson, Neusa Martins, Inka Bartsch.

**Data curation:** Gareth A. Pearson.

**Formal analysis:** Gareth A. Pearson, Neusa Martins, Pedro Madeira.

**Funding acquisition:** Gareth A. Pearson, Neusa Martins, Ester A. Serrão, Inka Bartsch.

**Investigation:** Neusa Martins.

**Methodology:** Gareth A. Pearson, Neusa Martins, Pedro Madeira.

**Project administration:** Gareth A. Pearson, Inka Bartsch.

**Supervision:** Gareth A. Pearson, Ester A. Serrão, Inka Bartsch.

**Visualization:** Gareth A. Pearson.

**Writing – original draft:** Gareth A. Pearson.

**Writing – review & editing:** Neusa Martins, Pedro Madeira, Ester A. Serrão, Inka Bartsch.

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
