## [Decision Letter · Decision Letter 0]

22 Jul 2019

PONE-D-19-18128

Sex-dependent and -independent transcriptional changes during haploid phase gametogenesis in the sugar kelp

Saccharina latissima

PLOS ONE

Dear Dr. Pearson,

Thank you for submitting your manuscript to PLOS ONE. After careful consideration, we feel that it has merit but does not fully meet PLOS ONE’s publication criteria as it currently stands. Therefore, we invite you to submit a revised version of the manuscript that addresses the points raised during the review process.

Both reviewers recommend publication of your manuscript, and suggest some minor improvements that you may want to consider in revising your manuscript.

We look forward to receiving your revised manuscript.

We would appreciate receiving your revised manuscript by Sep 05 2019 11:59PM. To enhance the reproducibility of your results, we recommend that if applicable you deposit your laboratory protocols in protocols.io, where a protocol can be assigned its own identifier (DOI) such that it can be cited independently in the future. For instructions see: http://journals.plos.org/plosone/s/submission-guidelines#loc-laboratory-protocols

We look forward to receiving your revised manuscript.

Kind regards,

O. Roger Anderson

Academic Editor

PLOS ONE

Journal Requirements:

2. We note that you are reporting an analysis of a microarray, next-generation sequencing, or deep sequencing data set. PLOS requires that authors comply with field-specific standards for preparation, recording, and deposition of data in repositories appropriate to their field. Please upload these data to a stable, public repository (such as ArrayExpress, Gene Expression Omnibus (GEO), DNA Data Bank of Japan (DDBJ), NCBI GenBank, NCBI Sequence Read Archive, or EMBL Nucleotide Sequence Database (ENA)). In your revised cover letter, please provide the relevant accession numbers that may be used to access these data. For a full list of recommended repositories, see http://journals.plos.org/plosone/s/data-availability#loc-omics or http://journals.plos.org/plosone/s/data-availability#loc-sequencing.

Additional Editor Comments:

Reviewer 1 has made some suggestions for clarification or expansion of some of your text. Please consider these minor recommendations when you revise your interesting manuscript.

Reviewer 2 also includes a few suggestions for your consideration.

Both appear to be constructive suggestions toward revision of your manuscript.

Reviewers' comments:

Reviewer's Responses to Questions

**Comments to the Author**

1. Is the manuscript technically sound, and do the data support the conclusions?

Reviewer #1: Yes

Reviewer #2: Yes

2. Has the statistical analysis been performed appropriately and rigorously? 

Reviewer #1: Yes

Reviewer #2: Yes

3. Have the authors made all data underlying the findings in their manuscript fully available?

Reviewer #1: Yes

Reviewer #2: Yes

4. Is the manuscript presented in an intelligible fashion and written in standard English?

Reviewer #1: Yes

Reviewer #2: Yes

5. Review Comments to the Author

Reviewer #1: The manuscript entitled “Sex-dependent and –independent transcriptional changes during haploid phase gametogenesis in the sugar kelp Saccharina latissima” by Pearson, Martins and colleagues overviews transcriptomics work to assess expression changes in male and female gametophytes during vegetative growth and induced gametogenesis. The manuscript is well-written, organized, and represents useful work on a less-studied life stage of a commercially important species. This reviewer has only a few comments that should be addressed prior to publication.

Major comment:

The manuscript lacks some discussion of future studies that can be taken to further address the questions raised. For instance, the authors state that transcription levels alone are limiting (Line 328), and parts of this work are preliminary. Please expand these points and provide new directions to investigate these topics. Would proteomics be important to clarify roles of the INO1 contigs? Could more detailed profiling of flagella-specific genes (using qPCR) be useful to differentiate male and female functions? Please expand on this in relevant discussion sections.

Minor comments:

Line 105. The authors state that vegetative growth conditions were based on initial experiments. Were these unpublished results, or were parameters based on previous growth trials from other studies, or in other kelp species? Please clarify.

Line 125. The reviewer is confused here with regards to biological replicates for sequencing. Combined sample weight is provided but not number of cultured samples (i.e., from multiple sporophytes). Were there at least three biological replicates used per sex per stage?

Line 125. Please provide a citation or procedure for the RNA extraction protocol.

Line 135. This reviewer understands the need to remove sequences from possible contaminants from the analysis, but could filtering for Stramenopile proteins remove important, uncharacterized sequences? If so, please address this point here.

Lines 194-195. Transcriptome shotgun assembly accession numbers GHNM00000000 and GHNM01000000 do not currently link to available data on NCBI (GenBank). This reviewers assumes that this is just not publically available yet. The authors should confirm this.

Line 304. The colon followed by a new sentence is confusing to this reviewer. Please rephrase.

Table 1 (Line 308). Why is there a line separating genes EIF4A and NRT? Please see similar situations in Tables 2 and 3. This is not clearly defined.

Line 324. Table S2 is listed twice here.

Lines 402-403. Please add a colon after “KEGG gene information.”

Lines 471-480. Please offer some clarification regarding the differential levels of the INO1 contigs. Is there any information regarding the second contig (elevated in F, undetected in M) as a variant that produces a different protein? What was the relationship between these two contigs – was one an upstream fragment while the other downstream, and could this give the reader additional information to further assess the expression difference? This should be explained in more detail.

Reviewer #2: Referee’s comments on

Sex-dependent and -independent transcriptional changes during haploid phase gametogenesis in the sugar kelp Saccharina latissima

The manuscript investigates the molecular aspects of haplodiplontic gametogenesis of the sugar kelp, Saccharina latissimi in the gametophytic phase. The authors used transcriptomic data from four time points during the transition from vegetative growth to gametogenesis. This study is well conceived and will contribute to our understanding of life cycle development involving haplodiplontic stages in Stramenopiles.

The study design is sound, and methods used are appropriate for the set of objectives outlines in the manuscript. The manuscript is well written and organized. Overall the work is interesting and well deserving of publication in the journal.

Below are some general comments for the authors’ consideration.

1. Please consider discussing implications of the study in detail and future investigation that can be done in gametogenesis of Saccharina latissimi and/or other related species with similar life cycle

2. It would be nice if the authors can provide a simplified model for gametogenesis pathways based on previous published systems and this study’s findings

Figure 5 caption is the longest I have seen in any paper. Please consider revising.

6. PLOS authors have the option to publish the peer review history of their article (what does this mean?). If published, this will include your full peer review and any attached files.

Reviewer #1: No

Reviewer #2: No

---

## [Author Response · Author response to Decision Letter 0]

13 Aug 2019

PONE-D-19-18128

Sex-dependent and -independent transcriptional changes during haploid phase gametogenesis in the sugar kelp Saccharina latissima

PLOS ONE

Dear Dr O. Roger Anderson,

We are very grateful for the for timely and positive reviews of our manuscript PONE-D-19-18128.

We have now completed a revised version that we hope will be suitable for publication in PLoS ONE. Responses to each of the reviewers’ comments are provided below in blue text, while the original comments are shown in italic. We found the reviewers’ comments constructive and useful in preparing the revised version. In the very few cases where we disagree or were unable to find a way to implement suggestions, we have tried to explain our reasoning.

First with respect to data availability:

Response: We believe all data have been made publicly available without restriction, it wasn’t our intention to suggest otherwise. This has been modified in the resubmission.

Reviewers' comments:

Reviewer #1: The manuscript entitled “Sex-dependent and –independent transcriptional changes during haploid phase gametogenesis in the sugar kelp Saccharina latissima” by Pearson, Martins and colleagues overviews transcriptomics work to assess expression changes in male and female gametophytes during vegetative growth and induced gametogenesis. The manuscript is well-written, organized, and represents useful work on a less-studied life stage of a commercially important species. This reviewer has only a few comments that should be addressed prior to publication.

Major comment:

The manuscript lacks some discussion of future studies that can be taken to further address the questions raised. For instance, the authors state that transcription levels alone are limiting (Line 328), and parts of this work are preliminary. Please expand these points and provide new directions to investigate these topics. Would proteomics be important to clarify roles of the INO1 contigs? Could more detailed profiling of flagella-specific genes (using qPCR) be useful to differentiate male and female functions? Please expand on this in relevant discussion sections.

Response:

We have tried to address these points with some further discussion. Certain issues may be beyond the scope of the paper, e.g., the need for fully characterized male and female haploid genomes. Some genetic tools such as mutant lines and screening also seem difficult given the life cycle of kelps (large and complex sporophyte stage). 

However, we have suggested (point 1) below) that proteomic analysis in kelps may be fruitful in further characterizing gene expression responses, given the early and apparently central role of ribosome biogenesis and translation. We also suggest that screening for mutants impacting reproductive development in the model species Ectocarpus may provide general insights:

1) Line 361 “The central role for transcriptional and translational control in gametogenesis suggests that proteomic comparisons would complement RNAseq-based approaches. Another promising future direction might be mutant screens in the model brown algal system (such as Ectocarpus) to analyse phenotypes impaired in gamete formation.”

2) Please see response related to INO1 expression below (lines 519-24), where we suggest some future directions in relation to this, and other sex-specific gene expression.

3) Flagella-specific genes: Yes, it is likely that deeper sequencing (RNAseq or targeted qPCR approaches), perhaps focused on later timepoints (in which more/more mature gametangia) are present would further clarify sex-dependent flagella-related gene expression. It will be of primary importance to obtain male and female gametophyte genomes for accurate read mapping to explore alternate gene copies and/or splice variants. We have added a sentence to this effect on line 625:

“In future, it will be important to obtain male and female gametophyte genomes for deep RNAseq mapping to explore sex-specific transcriptional variation and/or post-transcriptional variation (e.g., alternative splice variants).”

Minor comments:

Line 105. The authors state that vegetative growth conditions were based on initial experiments. Were these unpublished results, or were parameters based on previous growth trials from other studies, or in other kelp species? Please clarify.

Response: Indeed, the vegetative irradiance growth conditions were established principally by the second author for S. latissima. We also adapted nutrient conditions from previous work in S. japonica (Zhang et al. 2008, J Applied Phycol). The section has been changed to better reflect this starting on line 104 as:

“The irradiance conditions chosen for vegetative gametophyte growth were optimal based on initial experiments indicating improved culture health (mortality, qualitative assessment of pigmentation) in RL compared to WL (pers. obs.). Nutrient conditions were adapted from [28]”.

Line 125. The reviewer is confused here with regards to biological replicates for sequencing. Combined sample weight is provided but not number of cultured samples (i.e., from multiple sporophytes). Were there at least three biological replicates used per sex per stage?

Response: Perhaps this was missed by or not entirely clear to the reviewer, but it is stated on lines 128-9 (in the revised manuscript) that approximately 50 mg FW gametophytes were extracted per sample (i.e., per strain, sex and time point). Actually, after checking, this value is less than was used, which was between 100-200 mg FW per sample. The text has been corrected to reflect this.

On line 110 it is explained that the 2 strains were used as biological replicates (i.e., 2 replicates per sex at each timepoint; 2 sexes x 4 timepoints x 2 replicate strains = 16 samples in total). 

Again, on line 167- we state “Samples from the two available strains of S. latissima (SLO - Oslofjord and SLS – Spitzbergen) were used as biological replicates to investigate transcriptome expression profiles in response to the factors “sex” (two levels; male and female [M and F]) and “time” (four levels; vegetative growth in RL [=day 0], and 1, 6 and 8 days following a transfer to WL to initiate gametogenesis).” Unfortunately, these 2 strains were the only biological material available to us. 

Line 125. Please provide a citation or procedure for the RNA extraction protocol.

Response: The citation was provided on line 129 (now ref [29]: Pearson et al. 2006).

Line 135. This reviewer understands the need to remove sequences from possible contaminants from the analysis, but could filtering for Stramenopile proteins remove important, uncharacterized sequences? If so, please address this point here.

Response: The reviewers’ point is well taken. In this case, we preferred to take a conservative approach, particularly in light of the potential for relatively closely-related stramenopile contaminants in the study (diatoms, oomycetes, labyrinthulids), which could have affected the interpretation of expression patterns. In addition, from a functional point of view, uncharacterized sequences would add little to biological understanding, and we chose to concentrate on well-characterized and/or homologous sequences already described from brown algae (primarily the genome sequence of Ectocarpus Ec32, on which considerable annotation efforts have been concentrated). 

Lines 194-195. Transcriptome shotgun assembly accession numbers GHNM00000000 and GHNM01000000 do not currently link to available data on NCBI (GenBank). This reviewers assumes that this is just not publically available yet. The authors should confirm this.

Response: All read data and the assembled transcriptome have been deposited under SRA accession PRJNA547989, available online at https://www.ncbi.nlm.nih.gov/sra/PRJNA547989.

The TSA records have now been updated in the “Data availability” subsection of the “Materials and Methods” line 216. We note that the citation text in the manuscript follows that recommended by NCBI SRA.

Line 304. The colon followed by a new sentence is confusing to this reviewer. Please rephrase.

Response: We agree. The colon has been replaced by a full stop with no change in meaning (line 331).

Table 1 (Line 308). Why is there a line separating genes EIF4A and NRT? Please see similar situations in Tables 2 and 3. This is not clearly defined.

Response: We thank the reviewer, indeed this was not clear. Table 1 shows both transcription- and translation-related genes and nitrogen metabolism-related genes, which are separated by the line mentioned. This has now been made clear in the Table 1 legend (line 340). In addition, we mistakenly used “Fold-change” which has been corrected to “Log2 fold-change” in the table and legend.

Table 2: We have removed the internal separating lines.

Table 3: In this case, the line separates KEGG-annotated genes from 2 further contigs annotated as “flagellar associated protein putative” against different Ectocarpus Ec32 proteins. We feel in this case that the line is required and that its use is clear, so we have left the Table unchanged. If the Editor does not agree, we will of course make any changes necessary.

Line 324. Table S2 is listed twice here.

Response: Corrected (line 356)

Lines 402-403. Please add a colon after “KEGG gene information.”

Response: Now added (line 447).

Lines 471-480. Please offer some clarification regarding the differential levels of the INO1 contigs. Is there any information regarding the second contig (elevated in F, undetected in M) as a variant that produces a different protein? What was the relationship between these two contigs – was one an upstream fragment while the other downstream, and could this give the reader additional information to further assess the expression difference? This should be explained in more detail.

Response: In order to clarify the reviewer’s concerns, we have added a supplementary file (S1 File) containing an alignment of available stramenopile INO1 proteins as well as the results of our phylogenetic analysis. The results indicate that the 2 divergent S. latissima INO1 proteins fall within the brown algal (Phaeophyceae) clade. We also added the following (lines 520-24):

“Amino acid translations indicated that both encode functional full-length INO1 gene copies, based on protein alignments and InterPro annotations. Furthermore, phylogenetic analysis of available stramenopile INO1 proteins placed both copies in the brown algal clade (S1 File). Given the level of sequence variation between S. latissima INO1 copies, it should in future be possible to verify these data from genomic DNA, and study their expression in more detail using qPCR.”

We further note that on the phylogeny (S1 File) the non sex-biased contig is sister to a homologous gene from the congeneric taxon S. japonica. The genome sequence for this species is derived from a male gametophyte, and it is therefore interesting that the female-specific contig has no apparent homologue in the haploid male S. japonica genome. However, as our observations are based purely on expression in our de novo reference transcriptome, we feel it is too early to speculate on possible sex-specific gene copies.

Reviewer #2: Referee’s comments on

Sex-dependent and -independent transcriptional changes during haploid phase gametogenesis in the sugar kelp Saccharina latissima

The manuscript investigates the molecular aspects of haplodiplontic gametogenesis of the sugar kelp, Saccharina latissimi in the gametophytic phase. The authors used transcriptomic data from four time points during the transition from vegetative growth to gametogenesis. This study is well conceived and will contribute to our understanding of life cycle development involving haplodiplontic stages in Stramenopiles.

The study design is sound, and methods used are appropriate for the set of objectives outlines in the manuscript. The manuscript is well written and organized. Overall the work is interesting and well deserving of publication in the journal.

Below are some general comments for the authors’ consideration.

1. Please consider discussing implications of the study in detail and future investigation that can be done in gametogenesis of Saccharina latissimi and/or other related species with similar life cycle

Response: Please see our responses to reviewer #1 above.

2. It would be nice if the authors can provide a simplified model for gametogenesis pathways based on previous published systems and this study’s findings

Response: This would indeed be nice. However, with respect, we feel that attempting a conceptual model for gametogenesis at this point may be a little premature and we would like to avoid undue speculation. 

Figure 5 caption is the longest I have seen in any paper. Please consider revising.

Response: While we do accept that the legend to Figure 5 is very long, we feel that the annotation information regarding differentially expressed genes is necessary and useful to include in the body of the paper, rather than, e.g., as supplementary information. We are unsure that there is a more concise way to present these data and would therefore prefer to leave the figure in its current form (unless the Editor disagrees).

We hope that the revised version is satisfactory.

With best regards,

Gareth Pearson (on behalf of the co-authors).

---

## [Editor Report · Decision Letter 1]

28 Aug 2019

Sex-dependent and -independent transcriptional changes during haploid phase gametogenesis in the sugar kelp

Saccharina latissima

PONE-D-19-18128R1

Dear Dr. Pearson,

We are pleased to inform you that your manuscript has been judged scientifically suitable for publication and will be formally accepted for publication once it complies with all outstanding technical requirements.

With kind regards,

O. Roger Anderson

Academic Editor

PLOS ONE

Additional Editor Comments (optional):

Thank you for your careful attention to the recommendations of the reviewers. I am recommending acceptance of your revised manuscript.

---

## [Editor Report · Acceptance letter]

4 Sep 2019

PONE-D-19-18128R1 

Sex-dependent and -independent transcriptional changes during haploid phase gametogenesis in the sugar kelp *Saccharina latissima*

Dear Dr. Pearson:

I am pleased to inform you that your manuscript has been deemed suitable for publication in PLOS ONE. Congratulations! Your manuscript is now with our production department. 

With kind regards,

on behalf of

Dr. O. Roger Anderson 

Academic Editor

PLOS ONE